Upregulation of NOXA by 10-Hydroxycamptothecin plays a key role in inducing fibroblasts apoptosis and reducing epidural fibrosis

Dai Jihang
Sun Yu
Yan Lianqi yanlianqi@126.com
Wang Jingcheng jingchengwyz@163.com
Li Xiaolei
He Jun
Department of Orthopedics, Clinical medical college of Yangzhou University, Orthopaedic Institute, Subei People’s Hospital of Jiangsu Province, Yangzhou University , Yang Zhou , China
Liu Jie
Electronic publication date: 2017 Jan 12
Publication date: 2017
Volume: 5
Electronic Location ID: e2858
Received 2016 Aug 22; Accepted 2016 Dec 3
Copyright: ©2017 Dai et al.
Copyright year: 2017
Copyright holder: Dai et al.
License: This is an open access article distributed under the terms of the Creative Commons Attribution License, which permits unrestricted use, distribution, reproduction and adaptation in any medium and for any purpose provided that it is properly attributed. For attribution, the original author(s), title, publication source (PeerJ) and either DOI or URL of the article must be cited.
License URL: https://creativecommons.org/licenses/by/4.0/

Keywords: Epidural fibrosis, 10-Hydroxycamptothecin, Fibroblast apoptosis, NOXA

Funding: National Natural Science Foundation of China 81271994 81301550 81371971 81501870 Jiangsu Provincial Six Talent Foundation 2015-WSN-108 2015-WSN-110 Jiangsu Province Ordinary University Graduate Student Practice Innovation Engineering SJLX15_0678 This work was supported by grants from the National Natural Science Foundation of China (Grants#81271994, 81301550, 81371971, 81501870). Jiangsu Provincial Six Talent Foundation (2015-WSN-108 and 2015-WSN-110). Jiangsu Province Ordinary University Graduate Student Practice Innovation Engineering (SJLX15_0678). The funders had no role in study design, data collection and analysis, decision to publish, or preparation of the manuscript.

==============================
The fibrosis that develops following laminectomy or discectomy often causes serious complications, and the proliferation of fibroblasts is thought to be the major cause of epidural fibrosis. 10-Hydroxycamptothecin (HCPT) has been proven to be efficient in preventing epidural fibrosis, but the exact mechanism is still unclear. NOXA is a significant regulator of cell apoptosis, which has been reported to be beneficial in the treatment of fibrosis. We performed a series of experiments, both in vitro and in vivo, to explore the intrinsic mechanism of HCPT that underlies the induction of apoptosis in fibroblasts, and also to investigate whether HCPT has positive effects on epidural fibrosis following laminectomy in rats. Fibroblasts were cultured in vitro and stimulated by varying concentrations of HCPT (0, 1, 2, 4 µg/ml) for various durations (0, 24, 48, 72 h); the effect of HCPT in inducing the apoptosis of fibroblasts was investigated via Western blots and TUNEL assay. Our results showed that HCPT could induce apoptosis in fibroblasts and up-regulate the expression of NOXA. Following the knockdown of NOXA in fibroblasts, the results of Western blot analysis showed that the level of apoptotic markers, such as cleaved-PARP and Bax, was decreased. The results from the TUNEL assay also showed a decreased rate of apoptosis in NOXA-knocked down fibroblasts. For the in vivo studies, we performed a laminectomy at the L1-L2 levels in rats and applied HCPT of different concentrations (0.2, 0.1, 0.05 mg/ml and saline) locally; the macroscopic histological assessment, hydroxyproline content analysis and histological staining were performed to evaluate the effect of HCPT on reducing epidural fibrosis. The TUNEL assay in epidural tissues showed that HCPT could obviously induce apoptosis in fibroblasts in a dose-dependent manner. Also, immunohistochemical staining showed that the expression of NOXA increased as the concentrations of HCPT increased. Our findings are the first to demonstrate that upregulation of NOXA by HCPT plays a key role in inducing fibroblast apoptosis and in reducing epidural fibrosis. These findings might provide a potential therapeutic target for preventing epidural fibrosis following laminectomy.

Introduction

Laminectomy for treating lumbar disc herniation and other lumbar disorders often results in the formation of epidural fibrosis, and the development of epidural fibrosis is the major contributor to postoperative morbidities such as persistent low back pain and disability (Songer et al., 1995). Recently, it was demonstrated histologically that epidural haematoma, epidural fat accumulation and muscle invasion at the laminectomy site plays an important role in the formation of dense epidural fibrosis (Sen et al., 2005). Additionally, the number of fibroblasts is considered as a parameter for determining the density of epidural fibrosis (Burton, 1991; Mirzai, Eminoglu & Orguc, 2006; Cekinmez et al., 2009).

The prevention of epidural fibrosis has been a subject of concern for many years. In the last century, various biological or nonbiological materials were sought to reduce epidural fibrosis (Lee et al., 1990; Preul et al., 2010; Abitbol et al., 1994), but all of these techniques were not without complications. Recently, many investigators have carried out studies to prevent epidural fibrosis via promoting fibroblast apoptosis, and some of them have achieved satisfactory results (Sun et al., 2015; Yang et al., 2016). 10-Hydroxycamptothecin (HCPT), a chemotherapeutic agent, is used as an anti-tumour agent due to its specific suppressive effect on the cell cycle to treat different types of cancer in the clinical setting (Beretta, Perego & Zunino, 2006; Wang et al., 2007). Our previous study showed that local application of 0.1 mg/ml HCPT could conspicuously reduce postoperative epidural fibrosis formation in a rat laminectomy model (Sun et al., 2008). Recently, HCPT has shown its apoptosis-inducing character in some cell types (Yuan et al., 2016; Cheng et al., 2016), which implicated that it might be useful in the prevention of epidural fibrosis through inducing the apoptosis of fibroblasts.

NOXA, a member of the BH3-only family that is known as a vital regulator of cell apoptosis, promotes apoptosis mainly via producing heterodimers with active Bcl-2-like proteins (Oda et al., 2000). It has been proven that higher levels of NOXA can induce apoptosis in many tumour cell lines (Qin et al., 2005). Recently, it has been reported that the loss of NOXA expression protected mouse fibroblasts from DNA damage-induced apoptosis (Villunger et al., 2003). Additionally, another study showed that upregulation of NOXA promoted apoptosis in NIH 3T3 cells (Hershko & Ginsberg, 2004). All of these research results suggest that NOXA might play a key role in the process of cell apoptosis.

In summary, since the current treatments for epidural fibrosis are not satisfactory, the development of targeted therapies is particularly important, especially those aimed at fibroblasts. Therefore, we investigated the effect of HCPT on fibroblast apoptosis and epidural fibrosis via regulating NOXA. We showed that HCPT could induce fibroblast apoptosis and reduce epidural fibrosis by upregulating NOXA expression.

Materials and Methods

Reagent

HCPT was purchased from Aladdin Biotechnology Co., Ltd (Shanghai, China). The purity of 10-hydroxycamptothecin was 98%.

Cell culture and treatment

Fibroblasts were obtained from epidural scar tissue isolated from rats that underwent reoperation laminectomies. The cells were cultured at 37°C under 5% CO2 in Dulbecco’s modified Eagle’s medium (DMEM, Gibco, Grand Island, NY), containing 15% foetal bovine serum (FBS; Gibco), 0.1 U/L penicillin and 50 µg/ml streptomycin (Gibco, CA, USA). Cells in the exponential growth phase between passages 3 and 6 were used for all experiments. The fibroblasts were seeded onto various dishes and cultured overnight until they reached approximately 60–80% density, and then the cells were washed with phosphate buffered saline and treated with HCPT at various concentrations (0, 1, 2, 4 µg/ml) and for various durations (0, 24, 48, 72 h).

Cell lentiviral infection

Lentiviral infection was used to achieve gene silencing. The target gene NOXA was contained in the lentiviral vectors, which were purchased from Shanghai Genechem Co., Ltd. (Genechem, China). Fibroblasts were treated according to the instructions. Then, successfully transfected cells were used in the experiments evaluating the treatment of HCPT, Western blot analysis and TUNEL assay.

Western blot analysis

Following treatment, the fibroblasts were harvested and lysed in RIPA buffer (Beyotime, Hangzhou, China) on ice. After the lysates were centrifuged in 4°C at 13,000 ×  g for 10 min, the supernatants were collected for Western blot analysis. The protein concentration was determined by the BCA Protein Assay Kit (Beyotime, Hangzhou, China). The proteins were separated by 6%–12% sodium dodecyl sulphate-polyacrylamide gel electrophoresis (SDS-PAGE) and transferred onto polyvinylidene difluoride membranes (Millipore, Bedford, MA). Following blocking with 5% skimmed milk in TBST for 2 h, the membranes were incubated with the appropriate primary and secondary antibodies successively according to the instructions. The primary antibodies used were anti-NOXA, anti-cleaved-caspase3, anti-cleaved-poly ADP-ribose polymerase (cleaved PARP), anti-caspase3, anti-PARP, anti-Bax, anti-Bcl-2 and anti-β-actin antibodies (Cell Signaling Technology, Beverly, MA, USA). The anti-mouse or anti-rabbit IgG were also purchased from Cell Signaling Technology.

TUNEL assay in fibroblasts

A TUNEL assay was performed to detect the apoptotic effect of HCPT on fibroblasts. The apoptotic rate of fibroblasts was detected using the TdT-mediated dUTP-biotin nick-end labelling (TUNEL) test system (KeyGEN, Nanjing, China). All of the operating steps were according to the manufacturer’s instructions. Following a brief staining procedure, the features of apoptosis were evaluated via a fluorescence microscope. TUNEL-stained fibroblasts were believed to be apoptotic, and the total number of fibroblasts was counted by the DAPI-staining method.

Animals

In all, 72 Sprague-Dawley young adult male rats (mean weight of 280 g), purchased from the experimental animal centre of Yangzhou University (Yangzhou, China) were used for this study. All rats received care in compliance with the principles of Laboratory Animal Care according to international recommendations, and the study was approved by the Animal Care and Research Committee of the Yangzhou University, China. The animals were randomly divided into four groups (18 rats per group) as follows: HCPT (0.2 mg/ml), HCPT (0.1 mg/ml), HCPT (0.05 mg/ml) or control (saline). The rats were allowed to acclimate to the environment for 1 week before the experiment.

Animal model and topical application of drugs

Laminectomy models were performed according to a previous study (Sun et al., 2007). Briefly, following satisfactory anaesthesia by intraperitoneal injection of 1% pentobarbital sodium (40 mg/kg body weight), the fur around the location of L1 and L2 was shaved and the exposed skin was sterilized. After exposing the fascia and the paraspinal muscles by a midline skin incision, the L1 vertebral plate was removed by rongeur forceps. Then, complete 5 × 2 mm areas of dura mater were exposed. All rats underwent a total L1 laminectomy.

After disinfection and haemostasis of the lumbar region, HCPT concentrations of 0.2, 0.1 and 0.05 mg/ml or saline were applied to the laminectomy areas with cotton pads (4 × 4 mm) for 5 min. The surrounding tissues were covered by wet gauze to avoid touching the agent. After the cotton pads were removed, the decorticated areas of the laminectomy site were immediately irrigated with saline to remove the remaining HCPT. After the above operations, the wounds were closed in layers.

Macroscopic assessment of epidural fibrosis

For macroscopic evaluation, six rats were randomly selected from each group 4 weeks after laminectomy. All rats were sacrificed by an overdose of 1% pentobarbital sodium via intravenous administration. Then, the laminectomy sites were reopened, double-blind trials were used to assess epidural fibrosis, and the amount of fibrosis was judged based on the Rydell classification (Rydell, 1970): grade 0, no adhesions were apparent around the dura mater; grade 1, weak adhesions appeared around the dura mater but were easily dissected; grade 2, moderate adhesions appeared around the dura mater and could be dissected with difficulty without disrupting the dura matter; grade 3, dense fibrous adhesions were firmly adherent to the dura mater and could not be dissected.

Hydroxyproline content (HPC) analysis

HPC analysis was performed to assess the amount of collagen in the scar tissue. After macroscopic evaluation, approximately 5 mg of wet-weight scar tissue was obtained from the decorticated areas for HPC analysis according to a previous study (Yan et al., 2010). Briefly, the samples were lyophilised, ground and hydrolysed with 6 mol/l HCl at 130°C for 12 h. Then, 1 ml hydroxyproline developer (β-dimethylaminobenzaldehyde solution) was added to the samples and standards. The absorbances were evaluated at 558 nm using a spectrophotometer. In the end, the HPC/mg scar tissue was calculated according to the standard curve constructed with serial concentrations of commercial hydroxyproline.

Histological analysis

Four weeks after laminectomy, another six rats were picked randomly, and histological analysis was performed. After intraperitoneal injection of 1% pentobarbital sodium (40 mg/kg body weight), all rats received intracardial perfusion with 4% paraformaldehyde. The whole L1 spinal column, including the surrounding muscles and epidural fibrotic tissue, was removed and immersed in 4% paraformaldehyde. After four days of decalcification by 10% buffered formalin, each specimen was further decalcified in an ethylenediamine tetraacetic acid (EDTA) and glycerol solution for 30 days, and then embedded in paraffin again. Successive 4-µm transverse sections were made through the L1 vertebra from the top to the bottom. Twelve continuous transversal sections of 4 µm each from the top to the bottom of the upright L1 vertebra were made. Six odd sections were stained by means of haematoxylin and eosin (HE), and the epidural fibrosis was assessed by light microscopy using a 40×  objective. At a magnification of 400× and with three fields of the laminectomy sites in each section, the fibroblast density was calculated by using Image Pro Plus 6.0.

Scoring system of epidural fibrosis and epidural cell density

To perform impersonal histopathological evaluation, an epidural fibrosis and epidural cell density scoring system was used. The degree of epidural fibrosis (HE × 40) and epidural cell density (HE × 400) was graded according to previous studies (He, Revel & Loty, 1995; Yildiz et al., 2007). The scores of epidural fibrosis (HE × 40) were as follows: Grade 0, there was no fibrosis scar tissue around the dura; Grade 1, thin or weak fibrous bands were found around the dura; Grade 2, continuous adherence was observed in less than two-thirds of the laminectomy defect; and Grade 3, the fibrotic scar tissue adherence is extensive and firm, and more than two-thirds of the laminectomy defect were adherent with fibrous bands, or the nerve roots were also adherent with the fibrosis scar tissue. The epidural cell density (HE × 400) was graded as follows: Grade 1, no more than 100 fibroblasts under 400× magnified visual field; Grade 2, the count of fibroblasts was between 100 to 150 under 400× magnified visual field; and Grade 3, over 150 fibroblasts per 400× field.

TUNEL assay in fibroblasts of epidural tissue

Fibroblast apoptosis in the epidural tissue sections was also identified using the TdT-mediated dUTP-biotin nick-end labelling (TUNEL) test system (KeyGEN, Nanjing, China) according to the manufacturer’s instructions. After staining, the apoptotic fibroblasts were detected under fluorescence microscopy. The final images were merged and analysed by Image Pro Plus 6.0.

Immunohistochemical analysis

The six remaining rats from each group were utilised, and the acquisition of sections was according to the histological analysis above. Following deparaffinization and rehydration through gradient ethanol solutions, the sections were incubated in citrate buffer to activate the antigenicity, and then they were exposed to 3% H2O2 to block endogenous peroxidase activity. The sections were washed with PBS three times, and then the primary antibody (anti-NOXA) was added and incubated at 4°C overnight. The sections were then washed with PBS and incubated with the secondary antibody (biotin-labelled goat anti-immunoglobulin G) at room temperature for 1.5 h. Following treatment with 3,3’-diaminobenzidine solution for 5 min at room temperature, the nuclei were counterstained with haematoxylin for 3 min. The samples were observed under a light microscope.

Statistical analysis

The data from our experiments were analysed using SPSS 19.0 statistical software. All data are expressed as the mean ± standard deviation (SD). Statistical significance was defined as a P value < 0.05.

Results

HCPT induced apoptotic cell death in fibroblasts

To determine the apoptotic effect of HCPT in rat fibroblasts, we treated the fibroblasts with various concentrations (0–4 µg/ml) of HCPT for 24 h. As shown in Fig. 1A, the results from Western blot analysis revealed that HCPT could increase the expression of cell apoptosis markers such as cleaved PARP and Bax, while it decreased the expression of Bcl-2, which was considered as an anti-apoptotic marker. Moreover, we found that the effect of HCPT on these markers was dose-dependent. To further confirm the apoptotic effect of HCPT on fibroblasts, morphological examinations were performed. As shown in Fig. 1B, few TUNEL-positive cells were detected in the control group (1.86% ± 1.85%). Following HCPT treatment, the percentages of TUNEL-positive cells at1 µg/ml, 2 µg/ml and 4 µg/ml were 14.94% ± 1.40%, 20.06% ± 2.64% and 28.26% ± 2.64%, respectively (Fig. 1C). Taken together, these results indicate that HCPT significantly induced apoptosis in fibroblasts.

Figure 1 HCPT induced fibroblasts apoptosis.

(A) Western blot analysis revealed that HPCT could induce the expression of cleaved PARP and Bax, and decreased the expression of Bcl-2, in a dose-dependent manner. The histogram are presented as the mean ± SD of three independent experiments. *P < 0.05 versus control group. (B) TUNEL assay shown that the apoptotic rate of fibroblasts was also increased in a dose-dependent manner. The fibroblast nuclei were stained in blue, and TUNEL-positive cells were shown in green, (C) and the results were shown the bar graph.

HCPT increased NOXA expression in fibroblasts

To confirm whether HCPT affected NOXA expression in fibroblasts, the fibroblasts were treated with 2 µg/ml HCPT for 24 h, 48 h and 72 h. Following HCPT treatment, the Western blot analysis showed that HCPT could increase NOXA expression in a time-dependent manner. What’s more, the expression of cell apoptosis markers such as cleaved-caspase3, cleaved PARP and Bax was also increased with the increased expression of NOXA (Fig. 2). The result of the Western blot analysis showed that the application of HCPT could upregulate NOXA expression in fibroblasts and could promote fibroblast apoptosis.

Figure 2 HCPT up-regulated NOXA expression.

Western blot analysis showed that HCPT increased the expression of NOXA, which was accompanied by increasing expression of cleaved caspase3, cleaved-PARP and Bax, and the decreasing expression of caspase3, PARP and Bcl-2, in a time-dependent manner. β-actin was used as a control. The histogram represents the mean ± SD of three independent experiments. *P < 0.05 versus control group.

The effect of NOXA on fibroblast apoptosis

To detect the exact effect of NOXA in fibroblast apoptosis, we used lentiviral infection to downregulate NOXA expression to confirm that NOXA is required for HCPT-induced apoptosis in fibroblasts. The levels of NOXA expression revealed from Western blot analysis showed that we knocked down NOXA in the fibroblast lines successfully. NOXA knockdown increased the expression of Bcl-2 and decreased the expression of cleaved PARP and Bax (Fig. 3A). Also, in accordance with the results above, the rate of fibroblast apoptosis that was detected by TUNEL assay was decreased in the NOXA knockdown cells. What is more, the increased expression of NOXA, cleaved PARP and Bax indicated by Western blot analysis and the increased apoptotic rate detected by TUNEL assay, which occurred after HCPT treatment, were partially decreased by NOXA knockdown (Figs. 3B and 3C). Taken together, these results indicate that NOXA played a crucial role in HCPT-induced fibroblast apoptosis.

Figure 3 The effect of NOXA on fibroblast apoptosis.

(A) Western blot was performed to test the expression of NOXA and the apoptotic markers (cleaved PARP and Bax) in NOXA knockdown fibroblasts applied with or without HCPT. β-actin was used as a loading control. Statistical analysis was performed to analyse the band intensities of NOXA, cleaved-PARP and Bax/Bcl-2. The data were from three independent experiments. *P < 0.05 versus the control group. (B) TUNEL assay was used to detect the apoptotic rate of fibroblasts following NOXA gene deletion in fibroblasts treated with or without HCPT, (C) and the data are shown in the bar graph.

Macroscopic evaluation of epidural fibrosis

To explore the effect of HCPT on epidural fibrosis, laminectomy models were performed and evaluated. The results of epidural fibrosis were detected via macroscopic observation and graded according to the Rydell classification; the analysis showed that Rydell grade 3 existed in all of the control group rats. Grade 0, 1 and 2 were found in the HCPT-treated groups (Table 1). In the control group, thick and extensive adhesions were found around the dura mater, and it was difficult to remove the adhesions. In the 0.05 and 0.1 mg/ml HCPT-treated groups, weak or moderate adhesions were found around the dura mater, and the adhesions were easily dissected. However, in the 0.2 mg/ml HCPT-treated group, there were few adhesions around the dura mater, and the adhesions were easily removed without bleeding.

Table 1 The table of the grade of epidural scar adhesion.

The grade of epidural scar adhesion through macroscopic evaluation in rats according to the Rydell standard.

Group	Grade	
	0	1	2	3	
HCPT(0.2 mg/ml)	4	2	0	0	
HCPT(0.1 mg/ml)	1	3	2	0	
HCPT(0.05 mg/ml)	0	1	3	2	
Saline(9 mg/ml)	0	0	0	6	
Notes.

Six rats were randomly selected from the HCPT-treated group of different concentration and control group.

Hydroxyproline content (HPC) analysis

HPC analysis was used to detect the amount of collagen in the scar tissue. As shown in Fig. 4, the HPC in the HCPT-treated groups of 0.2 mg/ml, 0.1 mg/ml, 0.05 mg/ml, and the saline-treated group, were 25.78 ± 1.96, 31.21 ± 1.93, 42.82 ± 3.44 and 55.05 ± 4.43 µg/mg, respectively. The HPC levels in the HCPT-treated groups were less than that in the control group (P < 0.05). And the decrease of HPC was in a dose-dependent manner.

Figure 4 The effect of HCPT on epidural collagen tissue in rats.

HPC was expressed as µg/mg. The amount of hydroxyproline was decreased with increasing concentrations of HCPT. *P < 0.05 compared with the HPC in control group.

The effect of HCPT on epidural fibrosis in rats

To detect the effect of HCPT on reducing epidural fibrosis, histopathological evaluation was performed according to the scoring system of epidural fibrosis and epidural cell density. As shown in Figs. 5 and 6, thick epidural fibrosis with extensive adhesions to dura mater was found in the laminectomy sites of the control group; also, a particularly large number of fibroblasts were found in the scar tissue. Moderate epidural fibrosis was observed and fibroblasts were decreased around the laminectomy sites in the 0.05 mg/ml and 0.1 mg/ml HCPT-treated group compared with the control group. In contrast, little epidural fibrosis with fewer fibroblasts was found in the laminectomy sites of the 0.2 mg/ml HCPT-treated group. All of these results showed that the degree of epidural fibrosis in the HCPT-treated group occurred in a dose-dependent manner (Fig. 7).

Figure 5 The effect of HCPT on epidural fibrosis in rats.

The representative photomicrographs of the epidural fibrotic tissues in each group. (A) The images show that the loose scar tissues (asterisk) without adherence to the dura mater (arrow) were found in 0.2 mg/ml HCPT-treated group (Grade 0). (B and C) Moderate scar adhesion with slight adherence to the dura mater was found in 0.1 mg/ml HCPT-treated group (Grade 1) and 0.05 mg/ml HCPT-treated (Grade 2) group. (D) Dense scar tissue with extensive and tight adherence to the dura mater (Grade 3) was found in the saline group. The sections were stained with haematoxylin and eosin, and the magnification was 40×. “S” represents spinal cord, and “L” represents laminectomy defect.

Figure 6 The effect of HCPT on fibroblast in epidural scar tissue in rats.

The representative photomicrographs of fibroblasts in epidural fibrotic tissues in each group. (A) 0.2 mg/ml HCPT-treated group (Grade 1), (B) 0.1 and (C) 0.05 mg/ml HCPT-treated groups (Grade 2), (D) control group (Grade3). The number of fibroblasts decreased with increasing concentrations of HCPT, which occurred in a dose-dependent manner.

Figure 7 The effect of HCPT on fibroblast counting in epidural fibrosis tissue.

Fibroblast number was expressed as the number per counting area. *P < 0.05 versus the control group.

HCPT induced fibroblast apoptosis of rats

To confirm whether HCPT could induce apoptosis in rat fibroblasts, the TUNEL assay was used. As shown in Fig. 8, few TUNEL-positive fibroblasts were found in the control group. After HCPT treatment, the number of TUNEL-positive fibroblasts was increased, which occurred in a dose-dependent manner.

Figure 8 The effect of HCPT on fibroblast apoptosis in rats.

(A) Representative photomicrographs of fibroblast apoptosis in rats. The number of apoptotic fibroblasts increased as the concentration of HCPT increased, which showed that HCPT could induce the apoptosis of fibroblasts in rats in a dose-dependent manner. (B) The rate of TUNEL-positive fibroblasts is expressed in the bar graph. *P < 0.05 versus the control group.

The effect of HCPT on NOXA expression in epidural tissue in rats

To confirm whether HCPT could affect NOXA expression in epidural tissue in rats, immunohistochemical staining was performed. As shown in Fig. 9, we found that HCPT could increase NOXA expression in the epidural tissue in rats. Moreover, with the increased concentration of HCPT, the expression of NOXA also gradually increased. All of these results suggested that HCPT treatment could increase NOXA expression in the epidural tissue of rats.

Figure 9 The effect of HCPT on NOXA expression in epidural tissue in rats.

As the dose of HCPT increased, the expression of NOXA increased. (A) 0.2 mg/ml HCPT-treated group, (B) 0.1 and (C) 0.05 mg/ml HCPT-treated groups, (D) control group.

Discussion

Extensive epidural fibrosis following laminectomy or discectomy often results in negative effects on patients. Unsatisfactory clinical outcomes include radiculopathy, persistent low back pain, disability and so on (Lee et al., 2006; Yildiz et al., 2007). Several important factors such as the degree of haemostasis during surgery, postoperative chronic inflammation and lumbar instability influence the formation and development of epidural fibrosis (Sandoval & Hernandez-Vaquero, 2008; Tao & Fan, 2009). All of these factors would promote the proliferation of fibroblasts, and finally cause the formation of epidural fibrosis (Sun et al., 2008). Thus, target research on fibroblasts was considered as a good approach to prevent epidural fibrosis.

Because of its target-specific DNA-damaging ability, HCPT has satisfactory effects on inhibiting the proliferation of many tumour cells, and has been used to treat many types of malignant tumours (Zunino & Pratesi, 2004; Ulukan & Swaan, 2002). However, as an anti-tumour agent, not only can HCPT inhibit fibroblast proliferation but also can have an inhibiting effect on the proliferation of other types of cells around the laminectomy sites, which may result in nerve root damage and disturbance of wound healing. Thus, it is reasonable to doubt whether local application of HCPT could induce epidural fibrosis suppression with little side effects.

Apoptosis, also called programmed cell death (PCD), is a process of automatic, gene-controlled cell death that occurs in multicellular organisms, which is an important mechanism to maintain homeostasis (Wyllie, Kerr & Currie, 1980). Previous reports have suggested that fibroblast hyperplasia plays an important role in the process of fibrosis, which can determine the amount of scarring postoperatively (Sun et al., 2014; Liu et al., 2010). It was recently reported that reducing fibrosis can be achieved via inducing fibroblast apoptosis. Tang et al. (2012) reported that inducing apoptosis in human tendon capsule fibroblasts could be a potential approach for reducing excessive postoperative scarring. Li et al. (2016) reported that inducing the apoptosis of fibroblasts could prevent intra-articular fibrosis after knee surgery in rabbits.

As an important mechanism of cell death, apoptosis generally occurs via extrinsic and intrinsic pathways, also known as receptor-mediated or mitochondrial-mediated, respectively. Previous studies have shown that the Bcl-2 family of proteins performs an enormous role in the regulation of cell apoptosis. Also, it was well known that many stimulating factors can induce the upregulation of NOXA through P53-dependent and P53-independent pathways, and thereby cause cell apoptosis (Oda et al., 2000). As a pro-apoptotic protein, NOXA shows weak pro-apoptotic ability on its own (Ploner, Kofler & Villunger, 2008). However, recent studies showed that NOXA has indirect pro-apoptotic functions to start the caspase cascade and induce cell apoptosis, one of the mechanisms of which was realized by the BH-3 domain of NOXA binding with Mcl-1(Naik et al., 2007). It was reported that the upregulation of NOXA by 5-aminoimidazole-4-carboxamide riboside (AICAR) plays an important role in Bax/Bak-dependent apoptosis in mouse fibroblasts (González-Gironès et al., 2013). Moreover, Schuler et al. (2003) found that induction of NOXA occurred in P53-triggered apoptosis. It is known that HCPT can induce fibroblast apoptosis, but the exact mechanism of HCPT in inducing fibroblast apoptosis still needed to be elucidated.

In this study, we found that application of HCPT could induce fibroblast apoptosis, which was verified by Western blot analysis of apoptotic proteins and TUNEL assay of HCPT-treated fibroblasts. What is more, the results of Western blot analysis showed that HCPT could significantly upregulate NOXA expression in fibroblasts, which was accompanied by the increased expression of cleaved caspase3, cleaved PARP and Bax. Moreover, following the downregulation of NOXA in fibroblasts using a lentiviral inhibitor, we found that the expression of cleaved PARP and Bax had invariably fallen. Furthermore, the upregulated expression of NOXA and cleaved PARP following HCPT treatment were partially attenuated by NOXA knockdown.

In the rat models, the TUNEL assay of epidural tissues showed that HCPT could induce fibroblast apoptosis. Also, immunohistochemical staining showed that the topical application of HCPT could upregulate the NOXA expression in fibroblasts of epidural tissue, which further indicated that NOXA was an apoptotic promoter in rat fibroblasts. Combined with the effect of NOXA on fibroblast apoptosis in vitro, our results show that HCPT could lead to fibroblast apoptosis by upregulating NOXA expression, which may be the main effect of HCPT in preventing epidural fibrosis.

However, the process of epidural fibrosis formation is very sophisticated, and the proliferation of fibroblasts may be one of many factors that are involved in the development of fibrosis. There were certain limitations in this research. Many factors, such as inflammatory reaction, lumbar instability and so on, affected the formation of epidural fibrosis. However, we did not perform relevant experiments on these issues. As an anti-cancer agent, the effect of local absorption of HCPT at laminectomy sites is still unknown, and the local application time, area and concentration of HCPT should be restricted strictly to avoid side effects. Moreover, in the current study, we only investigated the effect of NOXA on fibroblast apoptosis, and further investigation on the definite signal pathways of epidural fibrosis formation should be carried out in the future.

Conclusion

In summary, our study showed that HCPT could induce fibroblast apoptosis and reduce epidural fibrosis by upregulating NOXA expression, which may provide a potential therapeutic target for preventing epidural fibrosis after laminectomy.

Supplemental Information

Data S1 The raw data of CCK-8 assay, Tunel assay, H&E staining and immunohistochemistry staining

Full-length uncropped western blots were contained in this file, the PVDF membranes were cut into pieces before exposure.

Click here for additional data file.

We thank Miss Zhou Dan, Mr. Zhao Shuai, Mr. Chen Hui, Mr. Zhu Gengyao, Mr. Sun Zhongwei, and Mr. Wang Shuguang for their excellent work.

Additional Information and Declarations

Competing Interests

Author Contributions

Animal Ethics

Data Availability

The authors declare there are no competing interests.

Jihang Dai conceived and designed the experiments, performed the experiments, wrote the paper.

Yu Sun analyzed the data.

Lianqi Yan prepared figures and/or tables.

Jingcheng Wang contributed reagents/materials/analysis tools, reviewed drafts of the paper.

Xiaolei Li and Jun He performed the experiments.

The following information was supplied relating to ethical approvals (i.e., approving body and any reference numbers):

Animal Care and Research Committee of the Yangzhou University, China

The following information was supplied regarding data availability:

The raw data has been supplied as a Data S1.

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
