# Peer review of "Upregulation of NOXA by 10-Hydroxycamptothecin plays a key role in inducing fibroblasts apoptosis and reducing epidural fibrosis"

_PeerJ, doi:10.7717/peerj.2858_

## Round 0.1 · original submission · Major Revisions

All the reviewers found this work was of interest, but was poorly presented. Please respond to reviewer’s questions point by point, and pay special attention to English.

For Figure 4 title, what is “suramin”? “HPC was expressed as μg/mg”, what does it mean? How do the topical concentrations of “mg/mL” correlate to “ug/ml” in cultured cells? The first letter for Fig. 7 label should be capitalized.

Reviewer 1 ·

Basic reporting

The manuscript is well written, has important messages, and should be of great interest to the readers. However, there are still some issues need to be revised. In general, I have some comments and suggestions to the authors highlighted below.

Experimental design

1. Is NOXA-KO mice, especially specific to epidural site available?
2. What about applying NOXA-KO epidural cell lines or primary cells or primary cells with immortalization to your study?

Validity of the findings

Abstract
1. Should add influence of current research on future studies in this field.

Introduction
1. It is better adding a summary at the end of the Introduction part.

Results
1. Please add more descriptions to each part of your results, they are relative simple now.

Discussion
1. Please state the advantages and disadvantages of your current study
2. Please add the influence of your current study on future related studies.
3. Please write future directions

Additional comments

The manuscript is well written, has important messages, and should be of great interest to the readers. However, there are still some issues need to be revised. In general, I have some comments and suggestions to the authors.

Reviewer 2 ·

Basic reporting

see below

Experimental design

see below

Validity of the findings

see below

Additional comments

The authors demonstrated that HCPT is effective in reducing epidural fibrosis and investigated the mechanistic regulation of NOXA in mediating the downregulation of fibroblast apoptosis.
The study is relatively well designed in terms of including both in vitro and in vivo study, reporting both phenomenon and underlying regulatory pathway. Yet, there are several aspects that need to be revised.
1. Decreased fibroblast apoptosis is one possible explanation for HCPT-induced reduction of epidural fibrosis. The current study design and findings could not exclude other possibilities, e.g. repressed inflammation, which was reported to be important to the promotion of epidural fibrosis. The authors are encouraged to include other possible mechanisms in their discussion.
2. The authors should provide histopathological grades in Fig. 5 and 6.
3. The writing of this manuscript is poor, making it difficult for reviewers to follow. The authors need to be especially careful about the grammar and format of writing, e.g. space was missing at several places. They need help from a native speaker to edit their English writing.
4. Significant modification is needed, especially for abstract to improve the logic flow in writing. The authors need to clearly point out their study design, such as what the treatment groups are, and rats underwent laminectomy at L1-L2 levels.

Reviewer 3 ·

Basic reporting

In this paper, Dai et al. have investigated the upregulation of NOXA by 10-Hydroxycamptothecin, which could play a key role in inducing fibroblasts apoptosis and reducing epidural fibrosis in vivo and vitro.

Experimental design

The authors investigated part of the intrinsic mechanisms of HCPT in causing apoptosis of fibroblasts, and whether HCPT has positive effects on epidural fibrosis of the laminectomy in rats.

Validity of the findings

See General Comments for the Authors

Additional comments

In this paper, Dai et al. have investigated the upregulation of NOXA by 10-Hydroxycamptothecin, which could play a key role in inducing fibroblasts apoptosis and reducing epidural fibrosis in vivo and vitro.

Main points

1. It is well known that in the caspase family, caspase-3 plays an essential role. Once activated, caspase-3 performed a number of executioner functions, including the activation of a latent. It has been reported that caspase-3 is the most efficient processing enzyme for PARP. Please explain why did you choose cleaved-PARP rather than others biomarker as a definitive indicator?
2. Please explain the time and dosage of HPCT treatment in FIG3A. For there is no explanation in method. There has been an obvious change of apoptosis protein in time and dosage curve with HPCT treatment, while the change is not discernible in NC control of Fig3A.
3. Since NOXA knockdown increased BCL-2 and decreased cleaved-PARP as well as BAX levels, than the effect of NOXA on fibroblast apoptosis applied with or without HCPT should be calculated by HPTC (+) / (-) change ratio in NC control and shNOXA group of FIG3 A.
4. Purity of 10-Hydroxycamptothecin should be mentioned.
5. The language needs to be improved by a native speaker or a professional language editing service.
- line 81 - "10-Hydroxycamptothecin (HCPT), a an chemotherapeutic agent"
- line 83 - "types of cancer in clinic clinical"
- line 107 - " were seeded into various various dishes "
- line 149 - "areas of dura mater was were exposed"
- line 229 - " The date Data showed that HCPT could upregulated NOXA expression in fibroblasts."
- line 235 - "The Levels of NOXA expression of from western blot"
- line 305 - "that HCPT could be significantly"

6. These references could be added to the article and discussed.

González-Gironès DM1, Moncunill-Massaguer C, Iglesias-Serret D, Cosialls AM, Pérez-Perarnau A, Palmeri CM, Rubio-Patiño C, Villunger A, Pons G, Gil J. AICAR induces Bax/Bak-dependent apoptosis through upregulation of the BH3-only proteins Bim and Noxa in mouse embryonic fibroblasts. Apoptosis. 2013 Aug;18(8):1008-16. doi: 10.1007/s10495-013-0850-6.

Schuler M, Maurer U, Goldstein JC, Breitenbücher F, Hoffarth S, Waterhouse NJ, Green DR. p53 triggers apoptosis in oncogene-expressing fibroblasts by the induction of Noxa and mitochondrial Bax translocation. Cell Death Differ. 2003 Apr;10(4):451-60.

Reviewer 4 ·

Basic reporting

The present manuscript provided both in vitro and in vivo evidences that the 10-Hydroxycamptothecin (HCPT) can trigger increased fibroblast apoptosis and its underlying mechanisms may relate to up-regulate NOXA expression. It is an interesting study since it may give potential strategies when using HCPT to prevent epidural fibrosis after laminectomy. But there are some main points need to be improved and revised.

Experimental design

Please see the comments below

Validity of the findings

Please see the comments below

Additional comments

1. When you try to connect the HCPT induced apoptosis to the possible mechanisms, it is better to look at all the key players in the apoptosis pathways. Please describe or draw a figure to point out the NOXA roles in the apoptosis pathway. What are the upstream and downstream genes of NOXA? Are there any other alternative BH3-only genes except NOXA? How do you discuss the possible mechanisms why HCPT can upregulate NOXA and then induce more apoptotic cells? Why are you confident that NOXA is the key player or the only key player during the fibroblast apoptosis? If previous studies are not clearly mentioned, I suggest the authors provided more evidences, such as the expression of p53 and caspase-3. qPCR would be a better way to first screen the key players.
2. Figure 3A, the western blot results of control group are not as significant as Figure 1, why? How did you apply HCPT? Dosage and time?
3. Please improve the writing to follow the logic flow. For example, explain why did you test the hydroxyproline content? You didn’t mention anywhere in the manuscript.
4. Figure 4 and 7, put control in the first column and followed by increased dosage.
5. Pay attention to the many typos and grammar usages. Some sentences are difficult to understand. For example, line 309-310 is not a full sentence. “Besides, the upregulation expressions of NOXA and cleaved PARP after HCPT treatment, which were partially attenuated by NOXA knockdown.”

---

## Round 0.2 · Minor Revisions

Your revised manuscript has been improved; however, some questions need to be addressed, especially from Reviewer #2 and #3.

Reviewer 1 ·

Basic reporting

All of my concerns and questions have been solved adequately.

Experimental design

All of my concerns and questions have been solved adequately.

Validity of the findings

All of my concerns and questions have been solved adequately.

Reviewer 2 ·

Basic reporting

see below

Experimental design

see below

Validity of the findings

see below

Additional comments

The authors have found decreased fibroblast apoptosis as one possible explanation for HCPT-induced reduction of epidural fibrosis. The current study design and findings could not exclude other possibilities, e.g. repressed inflammation, which was reported to be important to the promotion of epidural fibrosis. Thus, HCPT-induced reduction of epidural fibrosis could also result from other mechanisms, e.g. repressed inflammation. Have the authors measured inflammation markers? If so, the authors should present this and discuss. If not, the authors are encouraged to discuss these possibilities as limitations of the current study.

Reviewer 3 ·

Basic reporting

see below

Experimental design

see below

Validity of the findings

see below

Additional comments

The authors demonstrated that HCPT is effective in reducing epidural fibrosis and investigated the mechanistic regulation of NOXA in mediating the down regulation of fibroblast apoptosis. The manuscript has important messages, and should be of great interest to the readers. In this article, the author seriously answers the comments of the reviewers and has made details changes. In general, the manuscripts need some minor revisions. It is better adding a marker at the western blot figs; cleaved-caspase3 and cleaved –parp should be compared with caspase3 and parp in fig 2a and fig3b.

---

## Round 0.3 · accepted · Accept

Congratulations! Looking forward to receiving your contributions in the future.